# Design and Analysis of Independently Adjustable Large In-Pipe Robot for Long-Distance Pipeline

**Wentao Zhao [1], Liang Zhang [1],\* and Jongwon Kim [2]**

1   School of Mechanical, Electrical and Information Engineering, Shandong University (Weihai), Weihai 264209, China; 961454326a@gmail.com

2   Department of Electromechanical Convergence Engineering, Korea University of Technology and Education, Chungcheongnam-do 31253, Korea; kamuiai@koreatech.ac.kr

\*   Correspondence: zhangliang@email.sdu.edu.cn; Tel.: +86-130-6118-7255

**Abstract:** Large oil and gas pipelines are prone to corrosion and leakage, so in-pipe inspection is necessary. In this article, we show a novel robot mechanism for long-distance pipeline inspection. The robot consists of three crawlers and electric putters, which can adjust their speed and radius independently. Independent adjustment and system self-checking of the robot are achieved through multiple sensors. To make the robot operate efficiently, we studied the influence of size parameters on the forces between the central body and crawler. Moreover, we investigated how to adjust the attitude of the robot through the differential speed of the three crawlers. Static and dynamic simulations of internal forces are presented. The primary experiments indicate that our robot can operate stably in a large steel pipe.

**Keywords:** large in-pipe robot; long-distance pipeline inspection; differential; crawler

## 1. Introduction

Pipelines are widely used due to their convenience and economy. As demand for oil and gas is increases across the globe, we can expect pipeline networks will increase in length in the near future. However, as the service life increases, problems with pipelines will inevitably arise, such as aging, cracks, corrosion or external damage—especially for large oil and natural gas pipelines.

Leak-detection methods based on detection of vibration are widely used because of their advantages, such as wide range, high-precision and good expandability. This system ensures safe operation of pipelines over a wide range [1,2]. However, the pipeline structure itself cannot be inspected until it is too late to maintain after leaking [3,4].

The in-pipe robot has been a growing interest by reducing operator intervention from a dangerous work environment and improving work accuracy and efficiency in enclosed space. After more than 20 years of development, in-pipe robot field is gradually becoming mature; meanwhile, the ILI (in-line Inspection) technology has made considerable progress.

In-pipe robots can be classified into nine different robotic systems, such as PIG, wheels, crawler, inchworm, helical, etc. [5,6]. We focus on wheeled, cylindrical and crawler types because they have more controllability over their movements and stronger stability than others. There have been many studies done on these types of in-pipe robots [7,8] as these small robots have excellent motion control, enabling them to pass t-tubes, elbows and obstructions. However, we hope to make a comprehensive and accurate NDE (non-destructive evaluation) with MFL (magnetic flux leakage testing) and UT (Ultrasonic Testing) equipment [9]. Because of the strength and weight of the materials involved, small in-pipe robots cannot simply be scaled up to accommodate large pipes.

There are many studies on large-scale in-pipe robots. Kim (2011) focused on designing an optimal pipe navigation mechanism with a driving unit to overcome the various situations encountered inside steel pipes ($\varphi$600–$\varphi$800) [10]. Mateos (2012) presented another concept. Their robot consisted of six wheeled-legs that support the structure along the center of the pipe which enables cleaning and sealing tools to work properly by rotating around the inner circumference of the pipe, thus covering the entire 3D in-pipe space ($\varphi$800–$\varphi$1000) [11,12].

Most large and long-distance steel pipelines are straight, and the curved parts are smooth with large-angle bending. If the bending angle is too small, it causes water-hammer and wastes much energy. Robots are making new demands in this environment. On one hand, robots must get rid of the cable (the cable increases the robot's running friction), work alone in oily long-distance pipes and store information. On the other hand, large pipelines have heavy manufacturing and operating costs. Once they start operation, they will be filled with high-pressure oil sources. Therefore, any means of inspection into the pipeline must be carried out under the condition of fluid empties, and comprehensive testing data should be obtained. Visual equipment alone—like CCD—for inspecting inside a tube cannot find the defects inside the steel structure, which is an uneconomical investment. The robot itself should be used as a platform for testing and processing information. Moreover, it can carry other testing equipment into the pipeline for inspection [13–16].

This study focuses on the design of a large-scale robot with novel track-driving modules and independent pantographs controlled by the pressure sensors, encoder and laser radar. Three track driving modules (crawlers) are radially connected with an interval of 120°. Each crawler structure can expand and contract independently by an electric putter. To make the robot run stably in the long-distance pipeline, we established a mechanical model and differential rotation model to active control the robot attitude. The ultimate goal is to enable an in-pipe robot to independently perform inspections in large steel oil and natural gas pipelines under uncontrolled conditions.

## 2. Overview of Robot

The mechanism involved here can be divided into three parts: the central body, pantograph bracket and crawler, as shown in Figure 1b. The 45.8-kg robot can change its dimensions from 950 to 1200 mm. Combined with the battery, it can run for more than 65 min at a speed of 5.3 m/min.

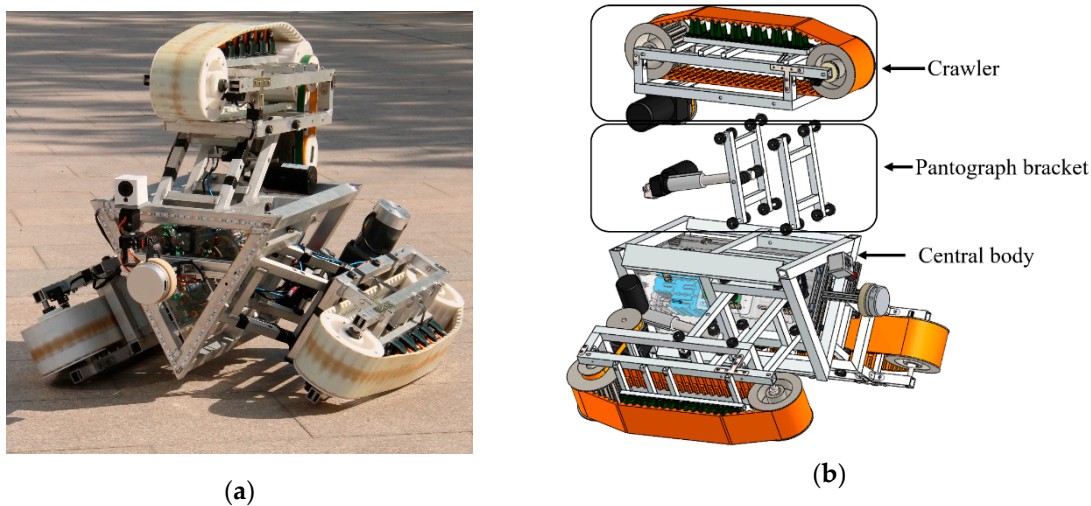

(**a**)  (**b**)

**Figure 1.** The large in-pipe robot. (**a**) Prototype. (**b**) Solid model.

*2.1. Design of Crawlers and Pantograph Bracket*

As shown in Figure 2a, instead of mounting the motor on the side of wheels, the gear motor was mounted on the upper of the tracked bracket to minimize crawler's width. Each 90-w DC motor was equipped with a speed reducer and powered with a voltage of 12 V.

When it comes to wall-press in-pipe robots, a flexible linkage mechanism allows the robot to adapt tp various pipeline diameters. Most of them use four suspension-links with one rod connect to an axial sliding bush. For some in-pipe robots, this saves space between the main body and the crawler section, but it brings many new problems. The thrust provided by the screw -ole or spring may not meet the expansion needs of large in-pipe robots. When locking phenomena happen, these parts are easy to damage. In the meantime, the robots may not easily maintain the state where the center of the main body coincides with the center of the pipe, because the actual diameter of the in-pipe robots will be slightly smaller than the pipe diameter. In this way to avoid possible self-locking phenomenon. However, the rugged inner surface of the tube will increase the vibration of the robot, and frequent vibration will cause hurting data performance. The robots and the pipe wall must maintain the proper pressure to avoid these two situations.

Therefore, the ideal support method can independently adjust the length of each "legs" and control the pressure between crawler and the pipe wall actively, which not only ensures the robot always in the center of the pipeline, but also reduces the harm caused by vibration and self-locking to the robot. As shown in Figure 2b, the pantograph consists of one electric putter and two parallel brackets. When the electric putter moves along the orange arrow, the crawler will rise or fall.

In addition, compared to the traditional screw structure, this design is easy to be replaced, robot can also quickly adjust its diameter according to the pipe diameter in a wide range (by adjusting the position of the connection point between the aluminum frame and the electric putter).

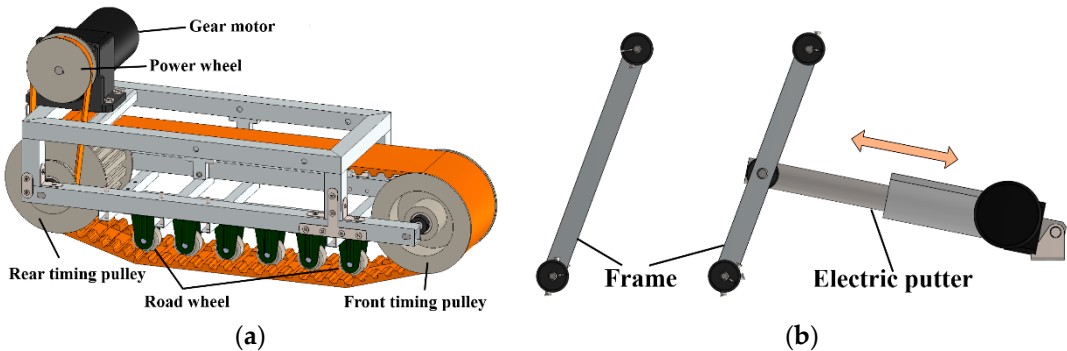

(**a**)            (**b**)

**Figure 2.** (**a**) CAD model of the crawler. (**b**) CAD model of the pantograph bracket.

*2.2. Detection System and Self-Checking System*

There are types of sensors installed in different parts of the robot, making it not only a mobile platform, but also as a kind of unmanned pipeline detector.

There are three types of sensors for motion control: pressure sensors, encoders and laser radar. As depicted in Figure 3, pressure sensors were installed between loading wheels and tracked bracket. Independent speed measurement and self-checking system depend on encoders. Abnormal conditions can be detected immediately. The laser radar was installed on an aluminum profile rod in front of the robot. As a crucial sensor of motion control, lidar can monitor the change of pipe diameter in real time. When deformation and siltation cause irregular terrain or meeting reducing coupling, elbows and *T*-branches, lidar makes an accurate judgment.

Getting stuck in pipe would be fatal for any robot. At present, most in-pipe robots are mainly based on artificial judgment and determining status by visual image. We want robot can have certain self-adjusting capabilities in complex environment, so we set up a self-checking system. Its main role is to ensure the safety of the robot's movement and will work in the cable or no cable operation.

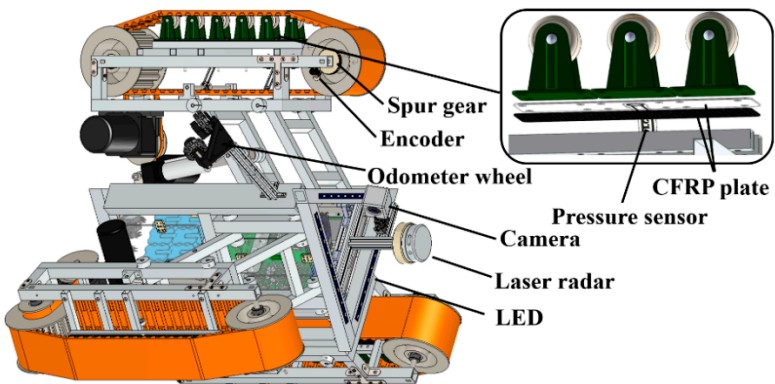

**Figure 3.** Sensors on the in-pipe robot.

First, a preliminary digital model of the pipe wall will be built by preliminary scanning with the infrared lidar, as shown in Figure 4. If there is an obstacle, the robot will judge how to pass the obstacle and react in advance. Second, pressure sensors in three crawlers sense the state of the robot. They will always ensure crawlers and the pipe wall are at the appropriate pressure. At the same time, when the pressure sensor perceives the pressure exceed the threshold value, it indicates that the robot may be stuck, and the electric push rod will make a small, dynamic contraction or relaxation until it returns to normal. The posture deflection, because of the robot imbalance or long-distance running, can be captured by the accelerometer. Three encoders on three crawlers will count the distance of each 'leg', and the three motors will adjust the speed independently to form a differential speed, so the robot returns to normal posture.

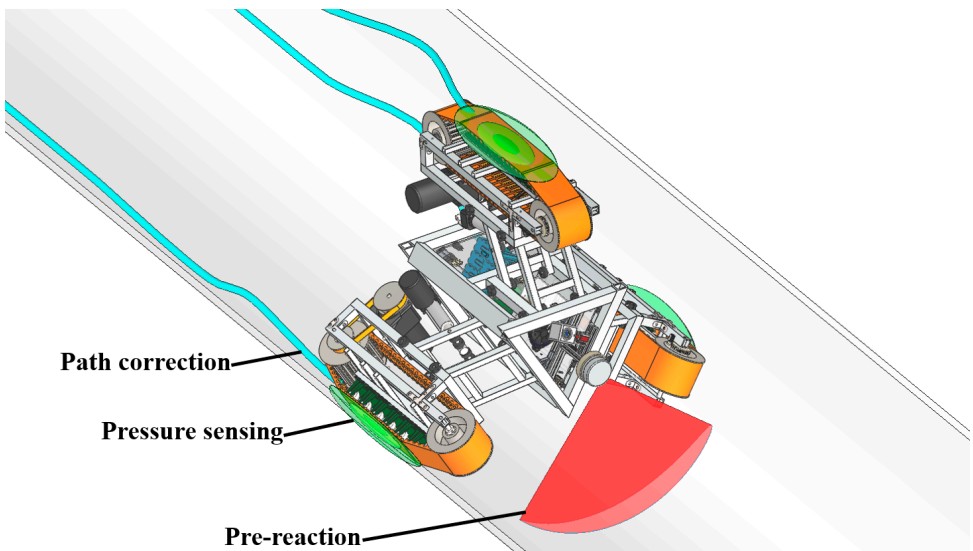

**Figure 4.** Self-checking system.

## 3. Long-Distance Stable Operation Model for In-Pipe Robot

### 3.1. Optimal Internal Force Scheme for Wall-Press In-Pipe Robot

Compared with wheel type, wall-press in-pipe robot has stronger active adjustment capability for different pipe diameters. To move over the muddy pipe internally, more researchers choose to combine the wall-press type with the track design [17–19]. Different from other core motor-driven telescopic designs, our robot's three "legs" (crawlers) controlled independently. Hence, we proposed a mechanical model of the rigid support between the crawler and the central body.

To simplify the calculation, we project force onto the symmetry plane of the crawler. The electric putter is simplified to a rigid light bar, and gravity is not counted except for the central body and crawler.

Where a, b, c, d, e, f, g, h, j, s are the external dimensions of robot, $\alpha$, $\beta$, $\gamma$ are the posture angle of the pantograph bracket. The dimensions are shown in Figure 5.

$$\begin{cases} F_1 sin\alpha + F_2 sin(\alpha + \theta) + F_3 sin\beta = G' \\ F_1 cos\alpha + F_2 cos(\alpha + \theta) = F_3 cos\beta \\ a{\cdot}F_1 sin\alpha + (a + g){\cdot}F_2 sin(\alpha + \theta) + (a + g + b){\cdot}F_3 sin\beta = G' j \end{cases} \tag{1}$$

where $\theta$ is the angle between $F_2$ and pantograph bracket, $G\prime$ is gravity of central body projected on this surface, $j$ is the distance from $O_1$ to $G'$.

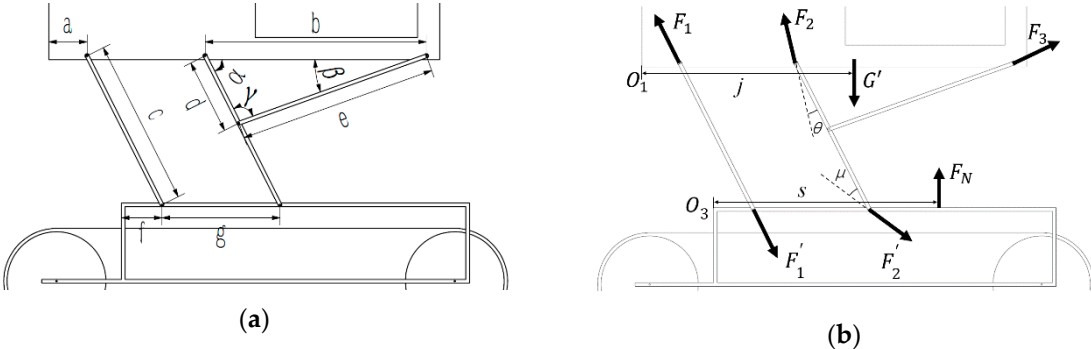

(a)           (b)

**Figure 5.** (**a**) Simplified model of robot frames with dimensions. (**b**) central body (origin is $O_1$) and crawler (origin is $O_3$ ) force distribution, $G'$ is the weight of central body, $F_N$ is the reaction force of $F_1'$ and $F_2'$.

Finally, we get the equation of forces acting on the crawler.

$$\begin{cases} F_1' sin\alpha + F_2' sin(\alpha - \mu) = F_N \\ f{\cdot}F_1' sin\alpha + (f + g)F_2' sin(\alpha - \mu) = F_N{\cdot}s \end{cases} \tag{2}$$

where $F_N$ indicates the equivalent force acting on crawler frame, $s$ indicates its distance from $O_3$.

The equation become solvable when the external dimensions of the robot are taken. Since the pantograph mechanism was calculated to make the crawler has effective support and to cope with the uneven pipe wall with different length electric putter, the robot is adaptable to the uneven pipe wall, as well as providing average supporting forces during movements.

### 3.2. Analysis of the Three-Crawlers Differential Rotation Model

When a large in-pipe robot is operating inside a pipe, the pipe is filled with tiny obstacles caused by corrosion and foreign matter. The robot has a certain route in straight pipeline, so robot path planning deviation exists [20]. Hence, actively adjusting the attitude to restore the robot to a stable attitude is necessary. Since the hardness of the obstacle cannot be detected, the strategy of the robot in the straight tube differs depending on the shape of the obstacle. To realize the control of differential speed rotation, we ought to establish its mechanical model base on force and pressure analysis.

Walking in a straight pipe for a long distance, the upper crawler should fit the wall, but no pressure ($F_a = 0$). Although more pressure means more driving force, reducing resistance is beneficial for an in-pipe robot to go farther when no other equipment is carried.

Where: $F_a$ , $F_b$ and $F_c$ denote the supporting force. $f_a$, $f_b$ and $f_c$ denote the friction. As in Figure 6a, when the robot is walking stable in straight pipe, assuming that the robot's weight is concentrated in the center and the constraint force is provided by the pipe wall. The following equation can be listed:

$$F_a + mg = (F_b + F_c)cos60 \tag{3}$$

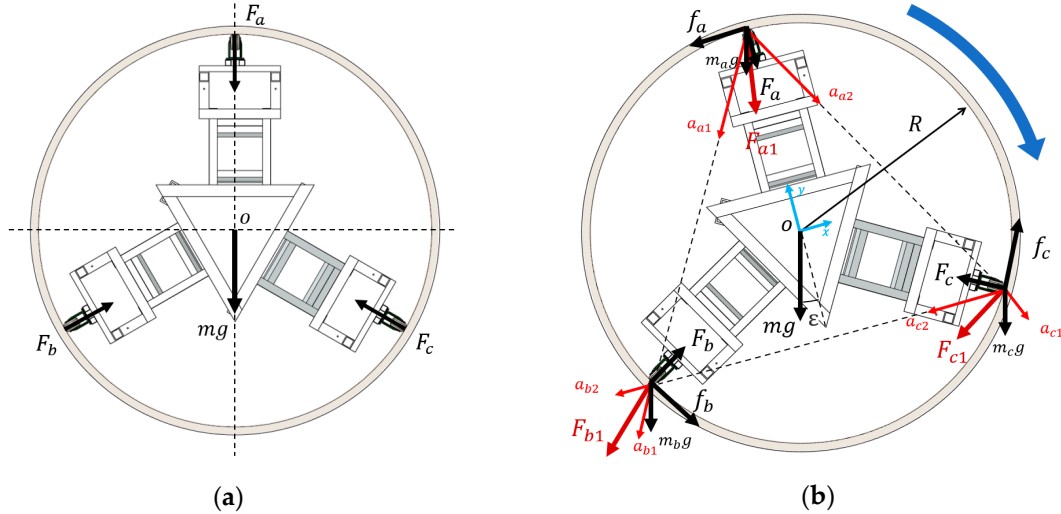

(a)             (b)

**Figure 6.** (**a**) Positive posture of the robot in the tube; (**b**) schematic of the force on the robot during differential motion (blue arrow represents the rotation direction).

However, when the attitude of the device is changed, the robot obtains a rotational movement through the differential speed and overcomes the dynamic friction.

Assuming the robot uses a three-crawler differential speed strategy to rotate inside the tube. We define that crawler has the radial velocity component $v_1$ and tangential velocity component $v_2$, which have the following relationship: $v_{a1} \geq v_{c1} \geq v_{b1}$, $v_{a2} = v_{b2} = v_{c2}$.

A Turning Center is formed between every two crawlers, and there is an imaginary turning plane between every two crawlers. The differential speed determines the position of the turning center on the plane.

Based on differential steering working principle, a steering center is formed between two different speed crawlers [21]. Meeting the following requirements:

$$r = \frac{b}{1 - \frac{v_b}{v_c}} \tag{4}$$

Figure 7b shows radial velocity between B crawler and A crawler, every two crawlers have a similar relationship. In particular, the fastest crawler (A) and the slowest one (B) has an opposite steering tendency with the other two groups (B and C, C and A).

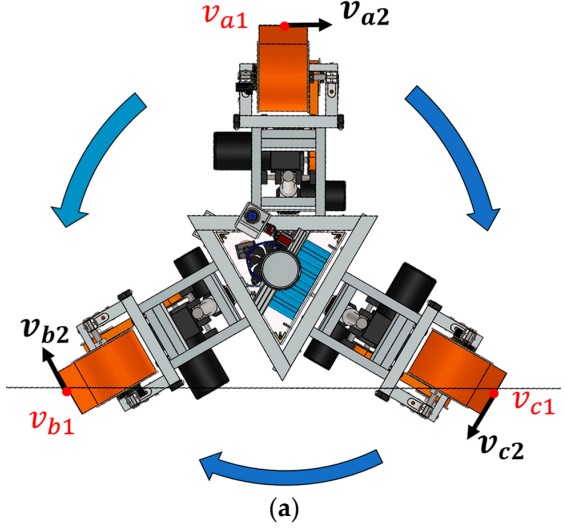

(a)

**Figure 7.** *Cont.*

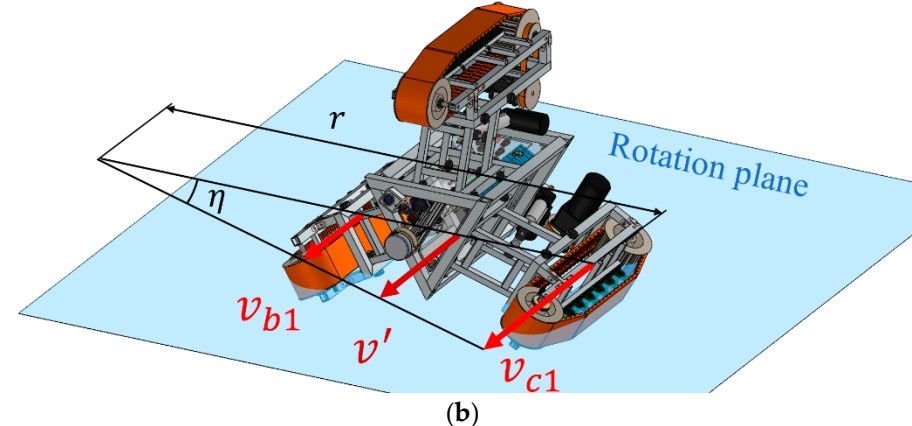

**(b)**

**Figure 7.** (**a**) Three-crawler differential speed strategy; (**b**) planar schematic of differential steering.

By using D' Alembert Theory, the force diagram is shown in Figure 6b. Coordinate system with robot, we established dynamic equation as follow:

$$\begin{cases} \vec{F}_{a1} = m_a \vec{a}_{a1} + m_a \vec{a}_{a2} \\ F_{a1y} + F_a + m_a g cos\varepsilon = 0 \\ F_{aix} - f_a - m_a g sin\varepsilon = 0 \end{cases} \tag{5}$$

$$\begin{cases} \vec{F}_{b1} = m_b \vec{a}_{b1} + m_b \vec{a}_{b2} \\ F_b cos60 + F_{b1x} - m_b g cos\varepsilon + f_b cos60 = 0 \\ f_b cos60 + F_b cos30 - m_b g sin\varepsilon + F_{b1y} = 0 \end{cases} \tag{6}$$

$$\begin{cases} \vec{F}_{c1} = m_c \vec{a}_{c1} + m_c \vec{a}_{c2} \\ f_c cos30 - F_{c1y} + F_c cos60 + m_c g cos\varepsilon = 0 \\ f_c cos60 - F_{c1x} - F_c sin30 - m_c g sin\varepsilon = 0 \end{cases} \tag{7}$$

The rotation of the robot is a dynamic process. $F_a$, $F_b$ and $F_c$, respectively denote the supporting force acting on crawlers, $f_a$, $f_b$ and $f_c$ denote the transverse friction between track and pipe wall, the mass of three crawlers and central body in the particle system are $m_a$, $m_b$, $m_c$ and $m$, respectively. Two equivalent centripetal accelerations ($a_{i1}, a_{i2}$) are generated on each crawler, respectively, which are equivalent to a force (fictitious inertial force) acting on the crawler. Its direction is perpendicular to the imaginary turning plane. There is centripetal acceleration in the rotation plane of every two crawlers, so each crawler has two sets of such accelerations ($a_{a1}$ and $a_{a2}$, $a_{b1}$ and $a_{b2}$, $a_{c1}$ and $a_{c2}$) and they synthesize a fictitious inertial force ($F_{ai}$, $F_{bi}$ and $F_{ci}$).

That is, at any moment in the movement of the robot, the main force, the binding force and the inertial force form a balance force system.

Then, the active roll moment and resistance moment of rotation can be written as:

$$\vec{M}_o(\vec{F}_{a1}) + \vec{M}_o(\vec{F}_{b1}) + \vec{M}_o(\vec{F}_{c1}) \geq (f_a + f_b + f_c)R \tag{8}$$

Since the different radial velocity $v_{a1}$, $v_{b1}$ and $v_{c1}$ mainly controlled by independent motors and supporting force on each crawler was got before, whether the rotation occurs and the influencing factors to tangential velocity $v_{a2}$, $v_{b2}$ and $v_{c2}$ are calculated. Therefore, we can actively control the posture of the robot by adjusting the speed of the three-crawlers.

*3.3. The In-Pipe Actively Adjusting Strategy*

When a wall-press in-pipe robot runs inside the pipe, we usually assume its geometric center coincides with the center of the pipeline. For small size in-pipe robots with flexible supports, this effect

can be ignored because the magnitude of the offset and vibration is small. However, for long-distance in-pipe inspection robot, offset is required to be controlled. On one hand, these errors will be amplified when running over long distances, causing the robot to gradually lose control, and eventually run to one side or become completely stuck. On the other hand, because large in-pipe robots often carry a variety of highly accurate sensors for inspection of the pipeline structure (such as infrared lidar and magnetic flux leakage modules), if the offset system error is not corrected, the sensor data processing would be trouble.

Establish the world coordinate system $x_w y_w z_w$ based on the pipeline and the $x_r y_r z_r$ coordinate system based on the in-pipe robot, as shown in Figure 8. Ideally, the robot should only have freedom of movement along the $z$-axis. However, in reality, many factors cause robot operation errors [12,22]. In $x_w o_w y_w$ plane, two translational degrees of freedom are affected by the difference in length between the three pantograph and the different deformation of the rubber track.

$$\vec{m_{xy}} = l_a + \vec{F_a}/k + l_b + \vec{F_b}/k + l_c + \vec{F_c}/k \tag{9}$$

where $\vec{m_{xy}}$ represents the offset from pipe center to robot center, $l_a$, $l_b$ and $l_c$ are the error of pantograph size, respectively. $k$ is the deformation coefficient of the rubber track.

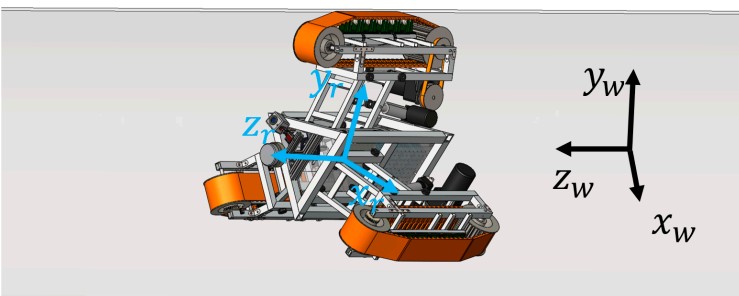

**Figure 8.** World coordinates and robot coordinates within the tube.

Due to different speed or different travel distances caused by the unevenness of the pipe surface, the robot rotates around the $y_w$. At the same time, the torque generated by differential speed mainly provides torque around the $z_w$. The torque around the $x_w$ comes from obstacles on the path and sudden acceleration and deceleration of the robot.

Figures 9–11 demonstrate solutions to problems encountered in pipeline by independently adjusting structure and differential speed regulation. Figure 9 shows the control method when the robot's attitude has deviated. In process ①, the robot gradually slides around the $z$-axis. Then, slide is detected by the accelerometer, the robot gradually returns to the normal attitude by adjusting three motors' speed, as process ② Correcting errors in long-distance pipeline operation is essential. Figures 10 and 11 show the solution when the robot encounters an obstacle in pipeline. In Figure 10, ① and ② indicates that the laser radar detects a sharp obstacle, robot moves backward. Next, repeating the previous active control process like Figure 9. Finally, resetting after passing the obstacle. When the obstacle is small or its gentle type, robot can directly lift one crawler over the obstacle as Figure 11 ②.

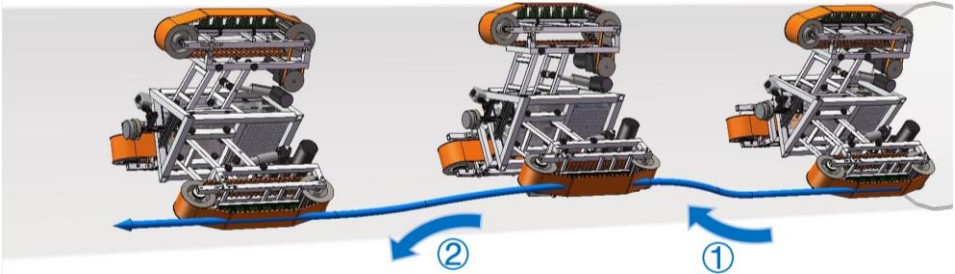

**Figure 9.** In-pipe robot positive-attitude control.

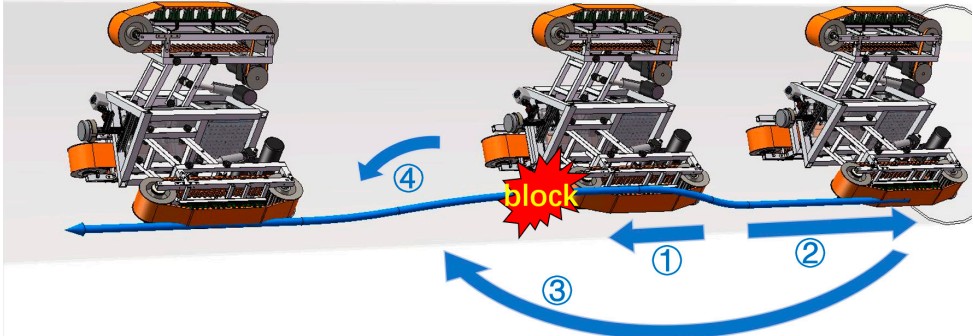

**Figure 10.** In-pipe robot passes indirectly through obstacles.

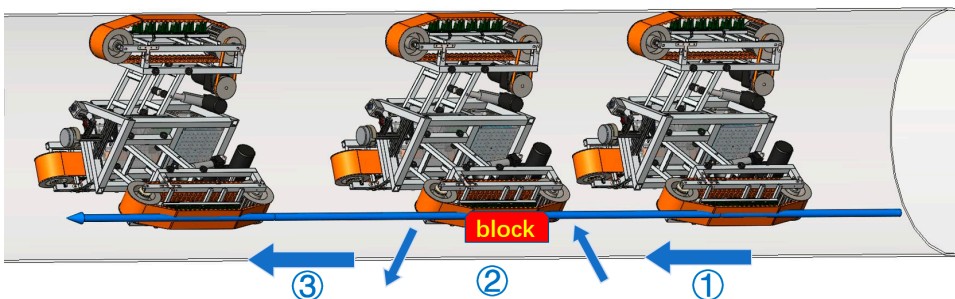

**Figure 11.** In-pipe robot passes directly through obstacles.

## 4. Simulation and Experimental test

### 4.1. Static Simulation of Internal Force Model

From the previously obtained Equations (1) and (2), we can analyze the force acting on the crawler to determine its size and whether it exceeds the pressure center. The previously defined parameter values are a = 0.052 m, b = 0.3085 m, c = 0.226 m, d = 0.1021, e $\in$ [0.200,0.290], f = 0.5504 m, g = 0.16 m, G′ = 112 N, j = 0.521 × 0.5 m. Based on the Taylor expansion, the $F_N$ and s were obtained by piecewise linear interpolation method. The result is shown in Figure 12 below.

As explained in Figure 12a, during length e increase, equivalent force $F_N$ rises rapidly and drops rapidly form a peak. Then increases slowly and almost linearly for the rest of the time, However, the arm of equivalent force position s suddenly decreases and then rises slowly. When the robot is in contraction state, as shown in Figure 12b, which the central body at the front of the crawler, the reaction force must be large enough due to the shorter arm. When the body and the crawler are closed in the axial direction, equivalent force $F_N$ and the arm of equivalent force s changed linearly.

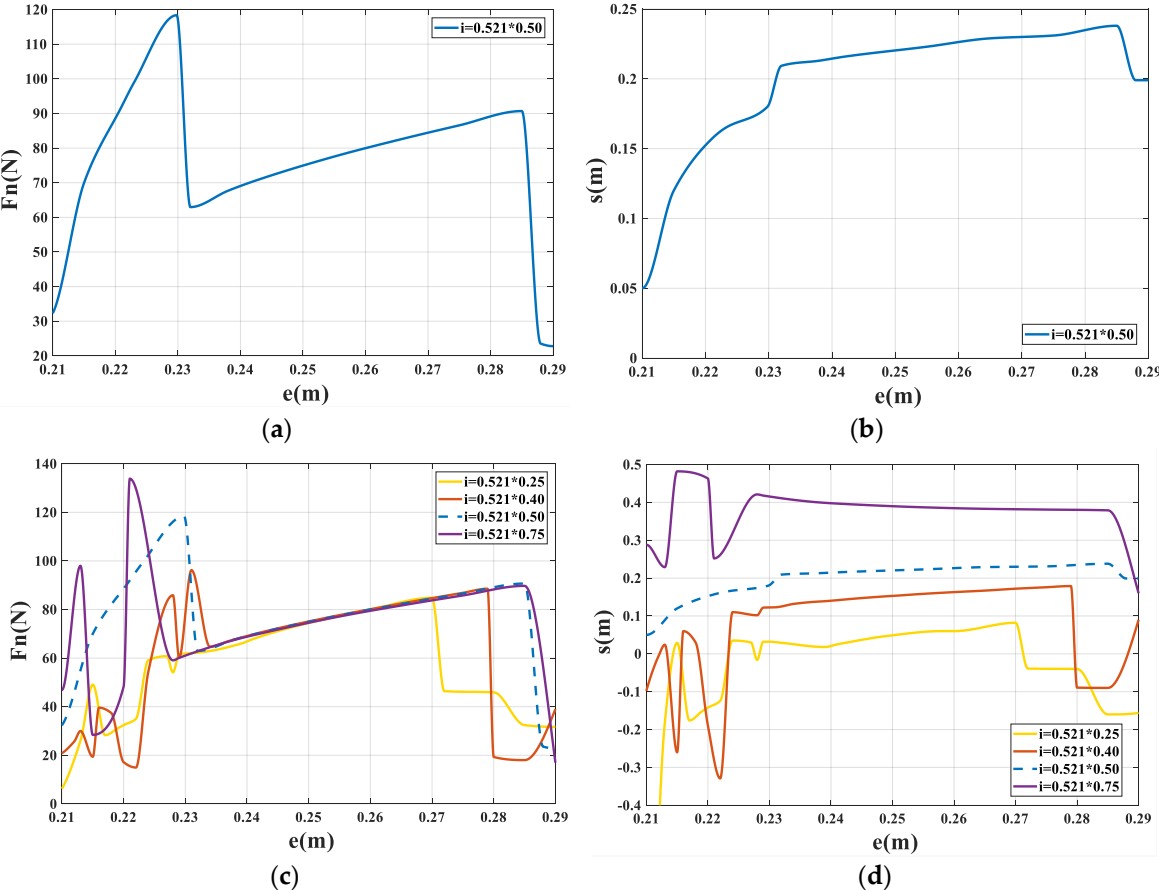

**Figure 12.** Changes in support force (**a**) and support force position (**b**) with the length of the putter e when the center of gravity is at the center of the robot (j = 0.521 × 0.5 m) and changes in support force (**c**) and support force position (**d**) with the length of the putter e (the center of gravity is not at the center of the robot).

In the actual situation, the position of the force could be different due to the position of the central body barycenter and crawler states. Figure 12c,d depict the variations of s and $F_N$ as the parameter j changes. The distance from $O_1$ to $G'$ (j) was set as 0.521 × 0.25, 0.521 × 0.40, 0.521 × 0.50 and 0.521 × 0.75 m. Compared to Figure 12a,b, the general trend has not changed and shows the same pattern. $F_N$ and s change with e and form linear zone. However, they fluctuate and shorter where the electric putter length e is shorter.

In addition, other meaningful parameters in the equation, such as the force of the putter $F_3$, the forces of the trusses on the central body $F_1F_2$ can obtain continuous effective solutions, which provide a reference for the robot design of similar institutions.

According to the ground pressure theory [23–25] the grounding pressure should be in the core area, so the range of s should be limited 0.024–0.329 m. To achieve the robot running stability in long-distance pipes, the length of the electric putter should be limited to the range allowed by the ground pressure center.

### 4.2. Dynamics Simulation

To verify the independent adjustment device and in-pipe robot differential motion process, we established simulation model by Adams.

First, a one-meter diameter experimental steel pipe was set. This pipe is provided with two gentle obstacles (up to 150 mm), located at the top and bottom left of the pipe, respectively. Each obstacle is 1.5 m long and two meters apart.

The robot was in a stable posture depicted in Figure 6a at the beginning. Gravity and friction were set close to the reality. Driving robot forward based on the force in each crawler. Set soft contact between robot and pipe.

By using two different ways, robot passed these two obstacles. The first was the traditional way of simultaneous expansion with three crawlers (red) and the second was the three-crawlers independent adjustment method (blue). As shown in Figure 13b, measured the pressure on the lower left crawler. Force on crawler in the independent adjustment device was more uniform and returned to stability faster.

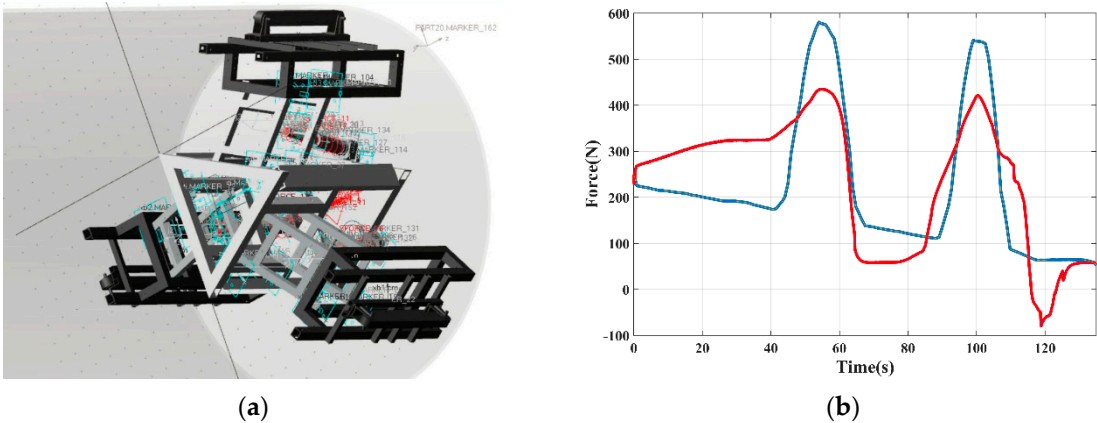

|     |     |
| :-: | :-: |
| (**a**) | (**b**) |

**Figure 13.** Independent adjustment analysis. (**a**) prototype in Adams; (**b**) feedback of the supporting force on one crawler in two ways.

Next, a one-meter diameter straight pipe is used for another simulation, which is just the same as our real steel pipe experimental platform. Other settings of the robot were the same as before.

Calculated by previous theory, the force provided by the motor (191.7 N) as the driving force. At the beginning, set the two crawler's force to increase or decrease by 1% as Figures 6b and 7a. Then, made the robot move forward until it stably runs. A virtual angle sensor was added to the robot to measure the rotation angle. In addition, the elapsed time was counted. After that, gradually increased the deviation of the force (PCT) and repeated the previous process. Maximum turning angle ($\varphi$) and the required time were obtained. As shown in Figure 14, with the increase of the force deviation, maximum stable angle and required time for stabilization gradually increased. Maximum turning angle robot could reach was 14°. In addition, robot motion would be very unstable when the differential ratio exceeds 15%.

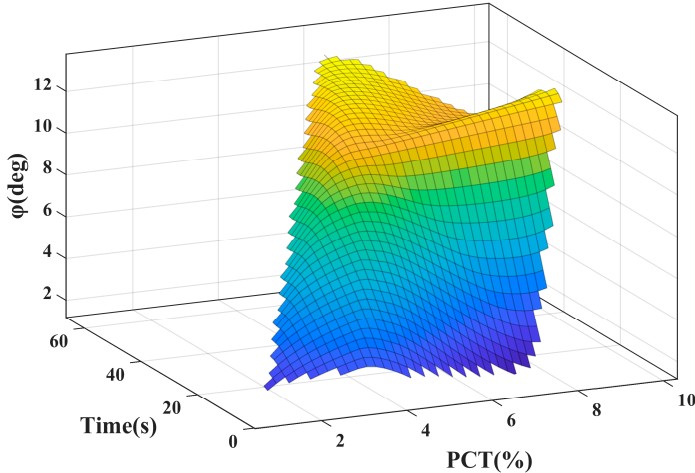

**Figure 14.** Rotation process time and maximum achievable angle under different tripod differential force ratios.

### 4.3. Experiment

To verify the fundamental capability of the large in-pipe robot, we made an experiment in a steel tube with an inside diameter of 990 mm. The robot ran repeatedly in three-meter pipes to verify our theory. Initially, the robot was placed into the pipe at the posture shown in Figure 6a, three motors were given the same power.

The top crawler was given a set pressure with the pipe wall. As shown in Figure 15, the robot completed the whole walking process in 23 s. Observing the motion of the robot, a slight offset had occurred in clockwise direction. This was not an ideal situation, so we monitored the data returned by the encoders and pressure sensors, as illustrated in Figure 16a,b.

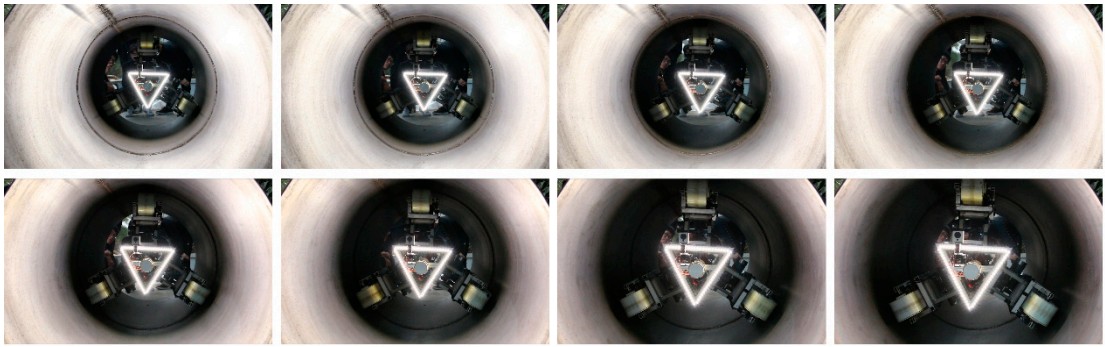

**Figure 15.** Straight driving experiment (every 3 s).

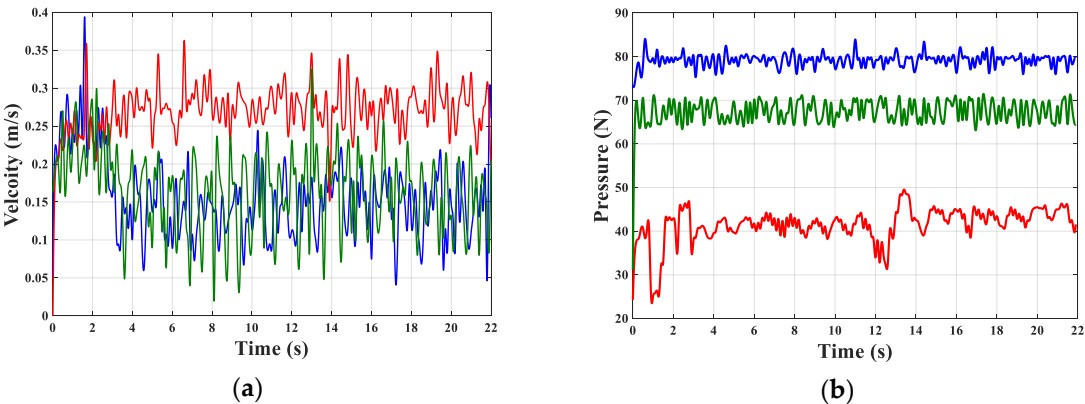

| (a) | (b) |
|:---:|:---:|

**Figure 16.** (**a**) Speed of three crawlers moving straight; (**b**) pressure on the three crawlers of moving straight.

Red, blue and green curves represented different crawlers, respectively. Blue and green curves represented the lower right and left crawlers, respectively. Since encoders directly measured the rotation of the track wheels, the measurement speed fluctuated greatly [26]. In general, speed of two lower crawlers is consistent, and the speed of the top track is the fastest. However, this was due to slippage, the actual distance is less than the measured distance. The pressure data showed different characteristics. The threshold value of the top red curve is given by us. The curve consisted of waveforms of pressure-putter adjustment process. The pressure on the lower two crawlers was within a uniform range, with no significant changing in speed at all. When the robot was designed, the structure of each crawler had the offset along the center, so there was a certain deviation in the pressure. Moreover, with Figures 6a and 7a, the offset was toward the crawler with higher pressure (blue), which is the same as the previous dynamic theory.

Another experiment was carried out and is shown in Figure 17. By changing the control strategy to three-crawlers speed consistency, the speed of every crawlers was adjusted by encoders, the robot had completed moving straightly in tube. According to the power measurement instrument, the

average power of the lower two crawlers were 56.5 w and 60.3 w. Meanwhile, infrared lidar scans the inside of the tube. The scan, shown as Figure 18, indicates that infrared lidar worked was stable and the vibration of the track part was not transmitted to the central body.

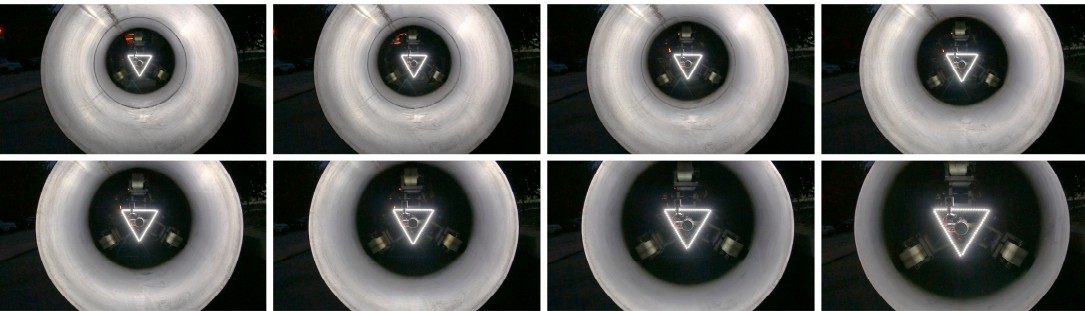

**Figure 17.** Straight driving experiment (every 3 s).

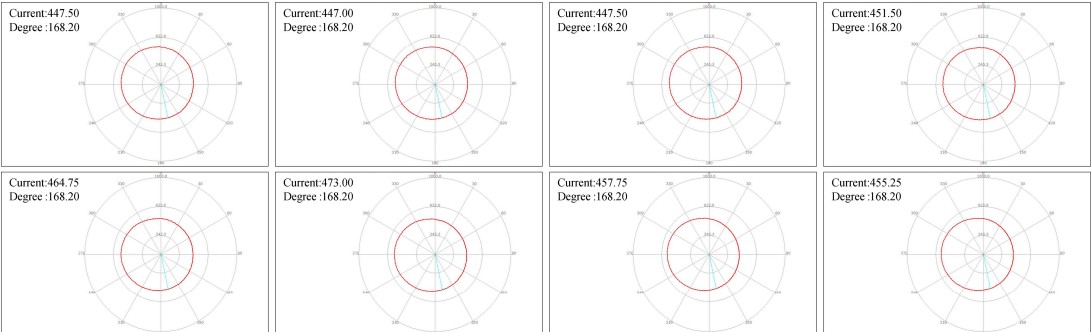

**Figure 18.** Infrared lidar test data (every 3 s).

Next, validated the theory of the in-pipe actively adjusting strategy. We used the PID algorithm to control the three-crawlers differential speed of the robot, as described in Figure 6b. The differential ratio is set to 3%. From Figure 19, the robot generated a rotation angle and reached its maximum rotation in 20 s and the robot completed the operation at this angle. Through many tests, the angle was about 10.9 degrees.

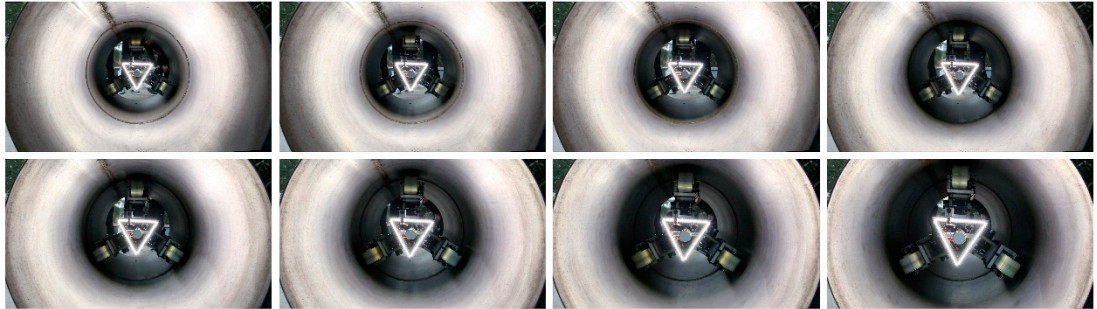

**Figure 19.** Differential speed driving experiment (every 3 s).

## 5. Conclusions

In this study, we presented a novel robot mechanism for large oil and gas pipe inspection. The robot's structure consists of three crawlers and electric putters, which are located circumferentially 120 degrees apart from each other. With the pressure sensors, encoders and laser radar, we can adjust their speed and radius independently. Internal forces for an in-pipe robot's independently adjusted device were analyzed. To ensure the robot's efficient operation in a long-distance pipeline, a three-crawler differential rotation model was built and an in-pipe active adjustment strategy was

proposed. The movement of the robot was verified in a $\varphi 1000$ mm tube. The experiment shows that the robot can change its attitude by adjusting the pressure and speed of each crawler.

In the future, methods for controlling attitude will be developed based on our mechanical model, which will enable the robot to autonomously walk in the long-distance pipeline without human control. The self-test system will respond flexibly to various situations in the pipeline by combining the data returned by various sensors.

**Author Contributions:** Conceptualization, L.Z. and J.K.; Methodology, L.Z.; Software, W.Z.; Validation, W.Z., L.Z.; Formal Analysis, W.Z.; Investigation, L.Z.; Resources, W.Z.; Data Curation, W.Z.; Writing-Original Draft Preparation, W.Z.; Writing-Review & Editing, J.K.; Visualization, J.K.; Supervision, L.Z.; Project Administration, L.Z.; Funding Acquisition, L.Z. All authors have read and agreed to the published version of the manuscript.

**Funding:** This research received no external funding.

**Conflicts of Interest:** The authors declare no conflict of interest.

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
