# Peer review of "Design and Analysis of Independently Adjustable Large In-Pipe Robot for Long-Distance Pipeline"

_applsci, doi:10.3390/app10103637_

Round 1

Reviewer 1 Report

The work is interesting but it is very important that the A.'s perform an accurate review of the English language, since it is sometimes difficult to understand few parts of the manuscript.

The paper is too long and the description of the dynamic model heavy and scarcely clear: also its validation and the correlation with the experiments is not straightforward.

Il it suggested that the paper is rewritten by putting in evidence the architecture of the system, its performance and the difference with the similar inspection systems already available on the market.

The A.'s should compare the performance of their crawler with respect to the ones already running and explain the main novelties introduced.

§2.1: it can be summarized but better explained: leaving alone few mistakes (V3 measured in "m" instead of "m/min", the word "normal vector" is used in place of "velocity", Z2=63.5 is a strange number of teeth, etc.), the A.'s do not explain the meaning of some variables nor they provide their values. 

§3.1: the whole section can be summarized:  equilibrium equations (6)-(9) are trivial therefore only the results of static analysis can be presented

§3.2: it is a difficult to understand section however the title is misleading, since it deals with static analysis. The next section is ill-referenced (3.2 again in place of 3.3)

§3.3: it is difficult to understand the modeling steps: once again it is better to write less equations and explain them more clearly, with appropriate figures.  Eq. (19) is wrong, unless in sliding conditions.

§4.1: parameter values are defined at the magnitudes of tenths of millimeters, which is unfeasible. It is also unclear if dynamic simulations have been run by means of Solidworks, which can be hardly reliable

§4.2: the tracked path of the experiments seem too short, for a 1 m large pipe.

Referenced literature is mainly taken from Chinese events: more international references should be consulted; ref. 17 is incomplete

Author Response

Response to Reviewer 1 Comments

Thank you very much for your positive evaluation of our paper. We are very grateful to the comments on our manuscript. According to your suggestions, we have tried to modify our paper as shown blow in details. If you have any other questions about this paper, I would quite appreciate it if you could let me know them in the earliest possible time.

Point 1:

  The work is interesting but it is very important that the A.'s perform an accurate review of the English language, since it is sometimes difficult to understand few parts of the manuscript.

  The paper is too long and the description of the dynamic model heavy and scarcely clear: also its validation and the correlation with the experiments is not straightforward.

   It suggested that the paper is rewritten by putting in evidence the architecture of the system, its performance and the difference with the similar inspection systems already available on the market.

Response 1: We re-checked the grammar using grammar software and removed redundant content. Reorganized the structure of the article and showed our innovations. Changed some descriptions in the previous section to enhance the association with the experiment

Point 2: The A.'s should compare the performance of their crawler with respect to the ones already running and explain the main novelties introduced.

Response 2:Reorganized the introduction, 2.1 and 2.2 of the article

Point 3: §2.1: it can be summarized but better explained: leaving alone few mistakes (V3 measured in "m" instead of "m/min", the word "normal vector" is used in place of "velocity", Z2=63.5 is a strange number of teeth, etc.), the A.'s do not explain the meaning of some variables nor they provide their values.

Response 3: Removed the introduction about the transmission part and focused on the independent adjustment part.

Point 4: §3.1: the whole section can be summarized:  equilibrium equations (6)-(9) are trivial therefore only the results of static analysis can be presented.

Response 4: Removed the tedious description and only kept the two formulas that are important for the static force analysis behind.

Point 5: §3.2: it is a difficult to understand section however the title is misleading, since it deals with static analysis. The next section is ill-referenced (3.2 again in place of 3.3)

Response 5: Completely removed the previous 3.2 and summarized the content in simple words at the back

Point 6: §3.3: it is difficult to understand the modeling steps: once again it is better to write less equations and explain them more clearly, with appropriate figures.  Eq. (19) is wrong, unless in sliding conditions.

Response 6: Removed unnecessary equations, added explanations of some parameters, and readjusted the order of interpretation of some equation parameters.

Point 7: §4.1: parameter values are defined at the magnitudes of tenths of millimeters, which is unfeasible. It is also unclear if dynamic simulations have been run by means of Solidworks, which can be hardly reliable

Response 7: Use adams to reset the original simulation (same settings as before) and add a new simulation.

Point 8: §4.2: the tracked path of the experiments seem too short, for a 1 m large pipe.

Response 8: Under the current conditions, the long-distance stability of the robot was verified by adding long-distance simulation in 4.1.

Point 9: Referenced literature is mainly taken from Chinese events: more international references should be consulted; ref. 17 is incomplete.

Response 9: Completed reference 17 and deleted some references from Chinese book.

Reviewer 2 Report

There have been many researches on pipeline inspection robots as described in the introduction. Therefore, the paper should focus more on explanation on the merits and demerits of the basic mechanical design and how it functions in real environment. How does the proposed robot move through T section or elbow section ? If the robot is designed for long distance pipeline, how can it be retrieved when the robot cannot return by itself ? I don't think there are many readers who are interested in gear ratio, force analysis, or kinematic analysis of this mechanism, because those results do not explain the novelty of the mechanism. I suggest reducing those sections and increase the later sections on simulation and experiment.

Author Response

Response to Reviewer 2 Comments

     We are very grateful to the comments on our manuscript. According to your suggestions, we have tried to modify our paper as shown blow in details. If you have any other questions about this paper, I would quite appreciate it if you could let me know them in the earliest possible time.

Point 1:

    There have been many researches on pipeline inspection robots as described in the introduction. Therefore, the paper should focus more on explanation on the merits and demerits of the basic mechanical design and how it functions in real environment. How does the proposed robot move through T section or elbow section? If the robot is designed for long distance pipeline, how can it be retrieved when the robot cannot return by itself? I don't think there are many readers who are interested in gear ratio, force analysis, or kinematic analysis of this mechanism, because those results do not explain the novelty of the mechanism. I suggest reducing those sections and increase the later sections on simulation and experiment.

Response 1:

     In the introduction, we changed the comparison to large in-pipe robots. In 2.1, we changed the content of the gear ratio to the introduction of independent supporting structure. In 2.2, we introduced how our self-checking system prevents the robot from jamming in pipeline. Most of the mechanics formulas in 3.1 have been deleted, the track analysis in 3.2 has been completely deleted, and the dynamic analysis in 3.3 has been simplified. Added a simulation in 4.2 and reworked the original simulation.

     Moreover, We re-checked the grammar using grammar software and removed redundant content. Reorganized the structure of the article and showed our innovations. Changed some descriptions in the previous section to enhance the association with the experiment

Round 2

Reviewer 1 Report

The English language must be thouroughly revised.

In eq.(5-10) is sometimes used the arrow to denote vectors with inconsistent notation. Eq. (8) is only valid in sliding conditions.

Apart from that, the manuscript is now far better than previous version and could be considered for publication.

Author Response

Response to Reviewer 1 Comments

Point 1:The English language must be thouroughly revised.

Response 1: Rewritten the abstract and introduction of the article. The grammar and sentences of some parts of the article have been completely revised, and the article has been revised using grammar software.

Point 2: In eq.(5-10) is sometimes used the arrow to denote vectors with inconsistent notation. Eq. (8) is only valid in sliding conditions.

Response 2: Fixed ambiguity due to subscript errors.      Fai→Fa1

Formula 8 has been deleted, and the friction force is explained in words
